# Establishment of Canine Oral Mucosal Melanoma Cell Lines and Their Xenogeneic Animal Models

**DOI:** 10.3390/cells13110992

**Published:** 2024-06-06

**Authors:** Shumin Li, Zichen Liu, Jinbao Lv, Di Lv, Huanming Xu, Hao Shi, Gang Liu, Degui Lin, Yipeng Jin

**Affiliations:** College of Veterinary Medicine, China Agricultural University, No. 2 Yuanmingyuan West Rd, Haidian District, Beijing 100193, China; b20203050416@cau.edu.cn (S.L.); lzc94@126.com (Z.L.); vet5216@163.com (J.L.); m15927308865@163.com (D.L.); 18810998613@163.com (H.X.); haoshi@cau.edu.cn (H.S.); liugang_0402@126.com (G.L.); csama@sina.com (D.L.)

**Keywords:** mucosal melanoma, cell lines, xenografts, live imaging

## Abstract

Canine oral melanoma is the most prevalent malignant tumor in dogs and has a poor prognosis due to its high aggressiveness and high metastasis and recurrence rates. More research is needed into its treatment and to understand its pathogenic factors. In this study, we isolated a canine oral mucosal melanoma (COMM) cell line designated as COMM6605, which has now been stably passaged for more than 100 generations, with a successful monoclonal assay and a cell multiplication time of 22.2 h. G-banded karyotype analysis of the COMM6605 cell line revealed an abnormal chromosome count ranging from 45 to 74, with the identification of a double-armed chromosome as the characteristic marker chromosome of this cell line. The oral intralingual and dorsal subcutaneous implantation models of BALB/c-nu mice were successfully established; Melan-A (MLANA), S100 beta protein (S100β), PNL2, tyrosinase-related protein 1 (TRP1), and tyrosinase-related protein 2 (TRP2) were stably expressed positively in the canine oral tumor sections, tumor cell lines, and tumor sections of tumor-bearing mice. Sublines COMM6605-Luc-EGFP and COMM6605-Cherry were established through lentiviral transfection, with COMM6605-Luc-EGFP co-expressing firefly luciferase (Luc) and enhanced green fluorescent protein (EGFP) and COMM6605-Cherry expressing the Cherry fluorescent protein gene. The COMM6605-Luc-EGFP fluorescent cell subline was injected via the tail vein and caused lung and lymph node metastasis, as detected by mouse live imaging, which can be used as an animal model to simulate the latter steps of hematogenous spread during tumor metastasis. The canine oral melanoma cell line COMM6605 and two sublines isolated and characterized in this study can offer a valuable model for studying mucosal melanoma.

## 1. Introduction

With the development of small animal clinical practice, diagnosing oral tumors in dogs is becoming more prevalent. Typically, dogs that develop oral tumors present with halitosis, salivation, oral bleeding, and difficulty eating, usually with rupture, mixed with microbial infections such as bacterial, viral, and fungal infections, leading to a decline in body condition. Melanoma is the most common malignant tumor found in the oral cavity in dogs [1,2]; it is highly localized and invasive, often metastasizing to local lymph nodes (LNs) and then to the lungs, and recurring after surgical resection, usually with a poor prognosis.

In recent years, some progress has been made in research on canine genomics [3,4,5] and transcriptomics analysis [6]. And the treatment of canine melanoma at the molecular level [7,8], immunotherapy [9,10,11], and other novel therapeutic approaches [12,13,14,15] have seen some progress. Still, their clinical effects and applications need to be further investigated. In addition, the causative factors and risk factors for canine oral melanoma are still unknown, such as the effect of oral inflammation on the tumor microenvironment and the link between periodontal disease and cancer [16].More clinical and research studies are needed on canine oral melanoma treatment and causative factors. Animal cell lines are still essential for studying tumors at this stage, and the number of established canine oral tumor cell lines is limited. The established cell lines have yet to be very well studied in animal models, and some cell lines have only been evaluated at the cell level [17,18,19]. Some early immunohistochemical (IHC) indicators are not consistently expressed from the primary tumor to the cell lines [20]. Studies on disease progression in canine melanoma have shown that metastasis is the most common cause of death, occurring in 80% of dogs [21,22,23]. Cell lines are the most commonly used cancer cells in metastatic models [24]. Modeling metastasis to study systemic therapy will be a focus of future research.

In comparative translational studies, the canine serves as a spontaneous model of mucosal melanoma, exhibiting a high degree of localized invasiveness and metastatic propensity, mirroring the biological characteristics of human mucosal melanoma. Similarities in genetic mutations and molecular pathways have been identified [25], making canine oral melanoma an optimal comparative and preclinical model for studying human mucosal melanoma [16,26,27,28,29]. Dogs, as the most prevalent companion animals, share a similar living environment with humans and are exposed to numerous common risk factors, so research on canine oral tumors will also contribute to advancements in human oral health.

We sought to enrich the canine oral tumor cell line library and provide a convenient and available research tool for the study of canine melanoma causative factors and treatment, as well as to benefit the study of human oral tumors. In this study, a canine oral mucosal melanoma (COMM) cell line, designated as COMM6605, along with two derived sublines, was stably passaged for over 100 generations. These cell lines were utilized to develop in situ implantation, subcutaneous implantation, and metastasis models in mice.

## 2. Materials and Methods

### 2.1. Primary Isolation and Passaging Culture

At the Teaching Animal Hospital of China Agricultural University, tumor tissue was surgically removed from a dog with oral tumors according to standard operating procedures with the owner’s knowledge and consent. The COMM6605 cell line was from a male Bichon Frise (medical record number 6605), and part of the tissue was paraffin-embedded, while the tumor type was diagnosed by pathological tissue interpretation. Additionally, fresh canine oral tumor tissue was sterilized by immersion in 75% alcohol for 1 min and then placed in a transfer fluid, Dulbecco’s modified Eagle’s medium (DMEM, 10567014, Gibco, GrandIsland, NY, USA) containing 1% penicillin–streptomycin (15140122, Gibco), 10 μg/mL gentamicin sulfate (G1914, Sigma-Aldrich, St. Louis, MO, USA), and 10 μg/mL ciprofloxacin (17850, Sigma-Aldrich).

Tumor tissues were then rinsed 5~8 times with PBS buffer solution (20012027, Gibco) containing penicillin 100 IU/mL and streptomycin 100 IU/mL, followed by the removal of superficial tissues and fibrous connective tissues with sterile scissors. Then, tissue fragments were cut into 1 mm-sized pieces with ophthalmic scissors in EP tubes. Tissue fragments were digested with DMEM containing type IV collagenase (C5138, Sigma-Aldrich, 0.5 g/L) and type I DNAase (90083, Thermo Fisher Scientific, Waltham, MA, USA, 0.2 g/L) in 60 mm diameter tissue culture dishes (Costar, Corning Incorporated, Corning, NY, USA) at 37 °C in a 5% CO_2_ cell incubator and incubated for digestion with shaking and mixing every 15 min. This was removed from the incubator every 45 min to 1 h and blown with a pipette gun. We collected the supernatant of the digested tissue mass, centrifuged it at 113× *g* for 5 min to collect the precipitate of digested tumor cells, filtered it with a 70 μm diameter cell filtration sieve (15-1070, Biologix, Jinan, Shandong, China), and then resuspended the cells in DMEM medium containing 10% fetal bovine serum (A5669701, Gibco) and 1% penicillin–streptomycin for 24 h. After 24 h of incubation and fluid exchange, the cells were left to grow to 80%~90% fusion for 3~4 d of incubation and then washed three times with PBS. After this, the cell lines were digested with 0.25% trypsin (25200072, Gibco) and then passaged for culture. The remaining undigested tissue blocks were reintroduced into the digestion solution and continued to be digested. After three rounds of digestion, most of the tissue mass was completely digested, and the remaining undigested tissue mass became loose; the remaining tissue mass was inverted and cultured in cell culture flasks, and then the flasks were turned over and continued to be cultured for 4~6 h when the tumor cells grew out of the tissue mass. Then, the cells were cultured for 1 wk, and the cells that grew out of the flasks were grown to 80%~90% fusion. The cells were then passaged. In addition, the cells were frozen every five generations.

### 2.2. Pathological and IHC Identification of Primary and Inoculated Tumors

The tumor tissues fixed with 10% formaldehyde were trimmed, dehydrated, and embedded to create tissue wax blocks. Then, they were sliced with a microtome and subjected to hematoxylin–eosin staining (HE) with histopathological interpretation. Based on the findings, the tissue underwent IHC staining followed by baking, dewaxing, antigen repair, and peroxidase blocking. Then, primary antibodies Melan-A (MLANA, ab234416, Abcam, Cambridge, MA, USA, 1 µg/mL), S100 beta protein (S100β, ab52642, Abcam, 1:500), PNL2 (ab12502, Abcam, 1:100), tyrosinase-related protein 1 (TRP1, ab178676, Abcam, 1:200) and tyrosinase-related protein 2 (TRP2, ab74073, Abcam, 1:100) were incubated overnight. After incubation of secondary antibodies (HRP Rabbit, 8114S, Cell Signaling Technology, Danvers, MA, USA; HRP Mouse, 8125P, Cell Signaling Technology) on the next day, DAB (P0203, Biologix) staining, hematoxylin recombinant staining, and dehydration transparency were performed. The process concluded with neutral gum sealing and microscopic examination for interpretation.

### 2.3. Growth Curve and Doubling Time of Cell Lines

We took the logarithmic growth phase cells, digested them with 0.25% trypsin, and adjusted the concentration of the cell suspension to 5 × 10^4^ cells/mL. We inoculated the sample into a 96-well plate with 100 μL/well. The cells were cultured in a cell culture incubator. Starting from the 24th hour, we mixed CCK-8 reagent (Cell Counting Kit-8, Beyotime, Beijing, China) with the complete medium at a ratio of 1:10, returned the medium to the 96-well plate, incubated it at 37 °C in darkness for 1 h, and then used an enzyme marker (ELx808™; BioTek, Agilent Technologies, Biotek, Winooski, VT, USA) to detect the optical density (OD) value at 450 nm for 7~10 consecutive days, with five replicate wells being detected every day. During this period, the cell culture medium was changed every 3 d. Non-linear regression was fitted using GraphPad Prism 8.0.2 (San Diego, CA, USA). The time required for the cells to double in value was calculated from the fitted equation.

### 2.4. Identification of Cell Line Species

We extracted the DNA from the cells, took an appropriate amount of DNA, and amplified the mitochondrial Cytochrome C oxidase subunit I (COI) gene of the animal with the germline fluorescent complex primer MIX. The amplified products were detected and analyzed by capillary electrophoresis on a gene sequencer (GenReader 7010, Beijing, China).

### 2.5. Clone Formation Assay

Stably passaged cells in the logarithmic growth phase were diluted to 500 cells/mL and inoculated into 6-well plates at 1 mL per well. The cells were incubated in a cell culture incubator with 5% CO_2_ at 37 °C for 7~10 d. Single-cell clone formation was then observed after staining with crystal violet.

### 2.6. Cell Immunofluorescence Staining 

We took logarithmic growth phase cells digested with trypsin and prepared cell suspension of 5 × 10^5^ cells/mL spread in 24-well plates with cell crawlers placed. When the cells grew to about 80%~90%, the crawlers were fixed with 4% paraformaldehyde and then permeabilized with 0.5% triton X-100 at room temperature and closed at room temperature for 30 min by adding drops of fetal bovine serum. Then, primary antibodies MLANA (1:100), S100β (1:100), PNL2 (1:100), TRP1 (1:100), and TRP2 (1:100) were added to the slides, and the slides were placed in incubation at 4 °C overnight; the next day, different fluorescent secondary antibodies were added to the slides respectively (goat anti-mouse IgG (H+L) AlexaFluor488, Invitrogen, Carlsbad, CA, USA, A-11029, 1:1000; goat anti-rabbit IgG (H+L) AlexaFluor488, Invitrogen, A-11008, 1:1000; goat anti-mouse IgG (H+L) AlexaFluor594, Invitrogen, A-11032, 1:1000; goat anti-rabbit IgG (H+L) AlexaFluor594, Invitrogen, A-11037, 1:1000) and dropwise DAPI (28718-90-3, Solarbio, Beijing, China) staining of the nuclei were performed for 5 min. Finally, the film was sealed with a sealing solution containing an anti-fluorescence quencher. The images were observed and captured under a fluorescence microscope.

### 2.7. Karyotyping

The vigorously growing cells were treated with 0.1 µg/mL colchicine to keep the chromosomes at metaphase; then, the chromosomes in the cells were exposed by cell fixation and sectioning, and finally, they were photographed and analyzed at the chromosome karyotyping workstation (Zeiss, Oberkochen, Germany) after staining by G-expressive banding.

### 2.8. Establishment of Orthotopic Xenograft and Subcutaneous Xenograft Models

At 4~6 wk old, female SPF-grade BALB/c-nu mice (SiPeiFu, Beijing, China) were selected and housed in the laboratory of the Animal Center of the China Agricultural University (CAU), and approval of experimental techniques was obtained from the Animal Ethics Committee of CAU.

Canine oral melanoma tumor cells grown in the logarithmic phase were selected and trypsin-digested, and single-cell suspensions were prepared at 2.5 × 10^7^ cells/mL. Suspensions of the cells were injected subcutaneously under aseptic conditions into the dorsum of each mouse, with 5 × 10^6^ cells in each 200 μL PBS suspension. The intra-oral injection site was the tongue. Each mouse was injected in the tongue with 40 μL of PBS suspension containing 1 × 10^6^ cells. For the control group, the same volume of PBS suspension was injected into exactly the same location. Each injection group consisted of five mice. Mice subjected to the tongue injection were anesthetized by intraperitoneal injection with 60 mg/kg of Zoletil^®^50 (Virbac, Carros, France), and the tumor cell suspension was injected when the mice showed good muscle relaxation, and the tongue could be pulled out of the mouth with tweezers.

The physiological status of the mice and the growth of the tumors were observed every 2~3 d, and the observation was continued until the end of the experiment. The length and width of the dorsal tumors were measured with vernier calipers every 2 d, and the weights of all mice were recorded. Mice were sacrificed by decapitation if they showed depression, a loss of appetite (rapid decrease by 20% of body weight), hypothermia (4~6 °C), or tumors more than 15 mm in diameter. Mice were sacrificed at the endpoint of the experiment, and their lungs and other organs were removed by dissection to observe tumor metastasis. Tumors and other organs were fixed in 10% formaldehyde and stained with HE for histological observation.

The volume of the tumor was calculated according to the formula:V = a × b^2^ × 1/2 (mm^3^)(1)
a and b represent the length and width of the tumor, respectively.

### 2.9. Establishment of Lentivirally Transfected Fluorescent Cell Sublines

A cell subline that expresses firefly luciferase (Luc) and enhanced green fluorescent protein (EGFP) genes named COMM6605-Luc-EGFP was established using lentivirus CV227 (Ubi-MCS-EGFP-SV40-firefly_Luciferase-IRES-Puromycin, GeneChem, Shanghai, China). Similarly, a cell subline named COMM6605-Cherry that expresses a red fluorescent protein (Cherry) was established using lentivirus GV298 (U6-MCS-Ubiquitin-Cherry-IRES-Puromycin, GeneChem). The lentivirus used in the article is a recombinant lentiviral vector, which is a pseudovirus based on HIV-1 (human immunodeficiency type I virus) and developed using the herpesvirus VSVG capsid protein, whose virulence genes have been eliminated and replaced by exogenous target genes. Lentiviruses can efficiently integrate exogenous genes into the host chromosome to achieve persistent expression. Pre-experiments were performed to determine the cellular MOI for lentiviral infection, the optimal infection conditions, and the working concentration of Puromycin for screening stably transfected cells.

Formal infection experiments were performed by preparing cell suspensions with 3~5 × 10^4^ cells/mL densities, using a complete medium to ensure that cell densities were around 90% at d 5. We replaced the medium with a complete medium containing lentiviral infection enhancer HitransG P (REVG003, Genechem) with an MOI of 50 for 12~16 h. Subsequently, the medium was replaced with a complete medium, and the culture was continued. It was possible to change the cells halfway through the incubation to maintain cellular activity. At about 72 h after infection, the infection efficiency was observed. Then, the stable transplants were screened with Puromycin (REVG1001, Genechem) for 48 h. Cells that were not successfully transfected after 48 h died, and the remaining cells were screened by reducing the concentration of Puromycin to the maintenance concentration (1/2~1/4 of the screening concentration) so as to continue the screening and amplification of the infected cells. The mixed clonal stable strain that was established was subjected to passaging culture and frozen for storage.

### 2.10. Establishment of Mouse Metastasis Model

Here, 1 × 10^6^ cells of COMM6605-Luc-EGFP were injected into 4-week-old BALB/c-nu mice (SiPeiFu) via the tail vein. Weight changes in the mice were recorded every 2~3 d, and the mice were imaged in vivo both 1 d and 1 wk after the injection, respectively, in order to trace the metastasis of the tumor cells in the mice. After the mice had been anesthetized using isoflurane anesthetics, 150 mg/kg of D-luciferin potassium salt (P0131, Beyotime) was injected intraperitoneally into the mice prior to imaging, and luminescence and EGFP were displayed on the IVIS imaging system (PerkinElmer, Waltham, MA, USA) after 5 min.

### 2.11. Statistical Analyses

A two-tailed unpaired t-test was used to evaluate the differences between the two groups. *p* < 0.05 was considered statistically significant. The results are expressed as mean ± SD and were analyzed using GraphPad Prism 8.0.2 (San Diego, CA, USA).

## 3. Results

### 3.1. Establishment of a Canine Oral Mucosal Melanoma Cell Line

A mucosal melanoma cell line COMM6605 was established from a canine oral tumor (Figure 1A). Under microscopic observation, cells cultured with tissue blocks were observed to crawl out from the tissue explants on day 3~5 of the culture period (Figure 1B). Isolated cell mixes are inoculated in cell plates, and visible intracellular melanin can be seen under light microscopy (Figure 1C). Cells obtained by the tissue block culturing method grew at a higher density and grew to fill the whole cell vial after 7~8 days of culturing for the first passaging. At this point, the cell line was stably passaged for more than 100 generations. Under the light microscope, the cells were spindle-shaped or polygonal, in a state of adherent growth (Figure 1D).

The cells’ growth curve is plotted using cells in the logarithmic phase (as shown in Figure 1E), and the cell multiplication time is calculated to be 22.2 h based on the fitted curve. After being propagated for over 100 generations in the laboratory, the cells are identified by species (Figure 1F). The COI gene was amplified by extracting DNA from COMM6605 cells, and the amplified product was detected and analyzed by capillary electrophoresis for definitive species identification. Mitochondrial protein-coding genes, such as the COI gene, are highly conserved and, therefore, are excellent targets for species identification. Many wild species have been accurately identified in recent years by labeling the first 648 base pairs of the COI gene [30,31,32,33]. The results for the COMM6605 cell line show specific amplification peaks at primer positions corresponding to the dog species, while no specific amplification peaks are observed at primer positions corresponding to other species. The result confirms that the cell samples are from a dog and are not contaminated with other species.

Histopathological identification of the canine oral tumor group show low magnification (Figure 2A) of the tissue, visible thickening of the oral mucosa, the junction of the mucosa and the submucosal layer containing a large number of tumor cells, convergent epithelioid growth, tumor cells with nested, sheeted, or striated distribution, some cells undergoing inter-cellular neoangiogenesis and tissue hemorrhage, and locally, the presence of inter-cellular dark brown granules. Under high magnification microscopy (Figure 2B), the tumor cells are heterogeneous, with unequal nuclei, prominent nucleoli, and multiple nucleoli and mitotic figures (more than 6/2.37 mm^2^). The cells contain a moderate amount of eosinophilic cytoplasm, and a few dark brown granules could be seen in the cytoplasm of the cells. The diagnosis was a histomorphology consistent with malignant melanoma.

Given the low pigmentation, IHC melanoma-specific indexes of the canine oral tumor tissue were assessed for further confirmation. The results of the test (Figure 3) were positive for MLANA, S100β, PNL2, TRP1, and TRP2, confirming that this was a malignant melanoma. Controls were identical canine tumor tissue controls without primary antibodies.

### 3.2. Characterization of the Cell Line

To characterize the COMM6605 cell line, we evaluated its proliferative capabilities, chromosome characteristics, and the expression levels of specific proteins.

We assessed the proliferative capacity of cells using single-cell clone formation assays. We inoculated low-density suspensions of single cells into a complete growth medium. After a 10-day incubation process, stained with crystal violet, cell colonies formed by single-cell clones could be observed with the naked eye. As shown in Figure 4, many single-cell clones are grown in all three replicate wells of the six-well plate. The cell community formed by each cell clone is a separate, purple-colored dot. The visualization shows the proliferation and viability of individual cells of this cell line. The successful establishment of these cell colonies indicates that the cell line has a high clonal potential.

Immunofluorescence staining of the cells shows that cells stably transmitted to 100 generations are positive for the specific IHC indexes MLANA, S100β, PNL2, TRP1, and TRP2 (Figure 5), which is consistent with the results of the IHC indexes of canine tumor tissues.

To evaluate the chromosome characteristics of the COMM6605 cell line, we conducted a karyotypic analysis of the tumor cells. One hundred cells with well-distributed and well-defined chromosomes were selected for observation and analysis to assess the number of chromosomes and the structure of the bulk karyotype of each cell. The standard dog karyotype comprises 38 pairs of acrocentric autosomes, a large sub-eccentric X chromosome, and small eccentric Y chromosome. COMM6605 cells appeared to be abnormal in both number and structure. Chromosome counting of 100 cells yields a chromosome number range of 45~74 for COMM6605 (Figure 6A), with a median chromosome number of 70 and chromosome deletions. The sex chromosomes consisted of XY, a male karyotype. A double-armed chromosome appeared as the marker chromosome of this cell line, originating from a mitotic fusion between chromosomes 1 and 5. As indicated by the red arrows in Figure 6B(1)~(3), these are marker chromosomes.

### 3.3. Successful Establishment of Orthotopic Xenograft and Subcutaneous Xenograft Models

To verify the tumorigenicity of the COMM6605 cell lines in mice, tumor cells were implanted into the tongue and dorsal subcutis of BALB/c-nu mice.

The tumor cell lines showed good tumorigenicity within the tongues of mice, resulting in marked enlargement and swelling of the tongue. (Figure 7A). Seeding of tumors on the tongue interfered with feeding as the tumors grew, resulting in significant weight loss on day 14 (Figure 7C). Mice were sacrificed to harvest the tongue tumors, showing significant swelling in the tongues of the tumor-implanted group compared with the control group (injected with the same volume of PBS) (Figure 7B). Mice inoculated subcutaneously had small lumps formed by the tumor cells visible at the inoculation site on day 2. The mass size grew until reaching the animal welfare and ethical boundaries on day 18 (Figure 7D), at which point the mice were sacrificed, and the dorsal subcutaneous masses were harvested (Figure 7E,F). There were no significant differences in the body weights of the mice in the tumor-implanted group during the implantation period compared to the control group (Figure 7G).

The tongue and dorsal masses underwent histopathological and IHC examination. The histopathological results for the tongue masses (Figure 8A) show that, under low magnification, the tongue mucosa is arranged in a papillary pattern. Tumor cells are present in the submucosal tissues, distributed in a sheet-like to river-like pattern, and are invasive. The dorsal masses (Figure 8B) are located in the subcutaneous tissue, with a nested or sheet-like distribution, and tumor cells are seen to invade the surrounding connective and muscle tissues. The morphologies of the tumor cells in the tongue and the dorsal masses are relatively similar under high magnification, with high heterogeneity, long spindle shapes, unequal nucleus sizes, pronounced nucleoli, and visible multinucleoli. The cells contain a large amount of eosinophilic cytoplasm, and brown granules are visible in the cytoplasm of some cells. The tumor-bearing mice were dissected to remove the lungs and other organs in order to observe metastasis of tongue and dorsal masses. The histopathological results show no metastatic foci in the lungs, liver, spleens, kidneys, or hearts (Appendix A).

Immunohistochemical staining of tongue swellings (Figure 9) and subcutaneous swelling on the backs (Figure 10) in mice show positive staining for MLANA, S100β, PNL2, TRP1, and TRP2, which are consistent with the immunohistochemical results of canine oral swellings and the COMM6015 cell line. The control group served as a negative control without a primary antibody, and its sections came from the same source as the experimental group, which were tumor tissues from the dorsal or lingual side of mice, respectively. Pathological interpretation and IHC results of the tumors on the dorsi and tongues of the mice show melanoma.

### 3.4. Successful Establishment of Sublines

Melanoma is a highly aggressive tumor with a high metastatic rate. In this study, we transfected the fluorescent protein gene and the luciferase gene in the COMM6605 cell line using two lentiviral vectors (see Section 2.9 in the Materials and Methods chapter for a description of these vectors) so that the tumor cells could be tracked after they were injected into the mice. We transfected the Luc and EGFP genes into cell line cells via lentivirus CV227 and established the COMM6605-Luc-EGFP cell subline via Puromycin screening. The expression of the EGFP fluorescent protein shows up green under the fluorescence microscope (Figure 11A). The Cherry fluorescent protein gene was transfected into the cells via lentivirus GV298, and the COMM6605-Cherry cell subline was established by Puromycin screening. The expression of the Cherry fluorescent protein was red under the fluorescence microscope (Figure 11B).

The biotin luminescence (Figure 12A) and fluorescence (Figure 12B) intensities observed in the COMM6605-Luc-EGFP cell subline under the IVIS imager are closely correlated with the cell inoculum quantity. The bioluminescence of the cells was imaged where the substrate D-luciferin potassium salt was added to the live cell culture medium. The fluorescence intensity of the Cherry protein in the COMM6605-Cherry cell subline is also found to be closely correlated with the cell inoculum amount (Figure 12C).

The COMM6605-Luc-EGFP cell subline was injected into BALB/c-nu mice via the tail vein, and significant bioluminescent signals were observed in the lung region by biotin luminescence imaging on day 6 after inoculation (Figure 12D).

## 4. Discussion

For the orthotopic xenograft model, the location of the implant in the oral cavity in this study was chosen to be intralingual. However, the most common lesion sites for canine oral tumors are in the gingiva and oral mucosa. In the context of oral tumor cell implantation in mice, tumor cells are usually transplanted onto the tongue [34,35], while tumor cells have also been implanted into the buccal mucosal tissue [36]. However, there is insufficient space in these parts of the mouse’s oral cavity during the actual inoculation, so we chose to place the implant intralingually, given our previous experience. The intralingual tumors implanted in this study still had some effect on the mice’s feeding. When the cancer had grown for about ten days, it caused mice to experience difficulty eating and drinking, resulting in weight loss. In this study, the mice were sacrificed when their weight loss approached 20%, and the tongue tumors were harvested.

Because the melanin expression in this cell line’s primary canine oral tumor is inherently low, in the mouse model with the established cell line, we also harvested tumors with only a tiny amount of brown pigmentation that did not exhibit a large amount of melanin deposition. For a more accurate characterization of the cell lines, we used a combination of several melanoma indicators in addition to the two indicators, MLANA and S100β, that are routinely recommended for clinical use. Several IHC antibodies can be used to help distinguish melanoma from other poorly differentiated tumors, and MLANA, PNL2, TRP-1, and TRP-2 are highly sensitive and 100% specific for the diagnosis of canine oral anaplastic melanocytic tumors [37]. A 2022 Oncology Pathology Working Group consensus on the most accurate methods for the diagnosis of canine melanoma cited the IHC markers MLANA, PNL2, TRP-1, and TRP-2 [38]. All of these are indicators that can be used for detecting the origin of melanocytes. The identification metrics used in this paper are a combination of test metrics recommended by the Pathology Working Group. A combination of several IHC indicators was used at all three levels: canine tumor tissue samples, isolated and established tumor cell lines, and in vivo tumor immunohistochemistry in xenografted mice.

The incidence of mucosal melanoma is higher in Asian populations. The etiology and pathogenesis are unknown [39]. Dogs are omnivores, like humans, and their living habits closely mimic those that produce the environment of the human oral cavity. They can thus be used to study the effects of pathogenic factors in the oral cavity on the occurrence and development of oral mucosal melanoma. Only a few canine oral mucosal melanomas have been established. Although cell lines cultured in vitro lose some of the cellular phenotypes and heterogeneity of the primary tumor by adapting to in vitro conditions and long-term culturing, they undergo extensive changes in gene expression, drug response, and cellular metabolism [40,41,42], which may limit their utility in modeling. Cell lines remain the most convenient research tool for mechanistic and metastatic studies [24], and the use of standard cell lines, replication experiments with multiple cell lines (similar key genotypes), and in vivo model supplementation can improve the credibility of cell line metastasis modeling studies. The cell lines established in this study stably expressed proteins of major specificity from primary tumors to cell lines and tumors grown in mice.

Mucosal melanoma is difficult to treat clinically because of its metastatic nature. Because most affected dogs eventually die from distal metastases, systemic therapy such as novel chemotherapeutic agents [43], targeted agents, and immunotherapy will be the focus of future research. Due to animal welfare and ethical considerations, our in vitro mouse model reached its endpoint in approximately two weeks, and no distal metastasis was detected in local inoculation due to species isolation. Subsequently, we established fluorescence-expressing cell lines via lentiviral transfection, metastasis models via in vivo mouse modeling, and the visualization of metastasis during disease progression via in vivo imaging; EGFP was poorly detected due to the depth of the body cavity, a phenomenon that has been seen in other established EGFP cell lines [44]. However, Luc bio-luminescence was effective due to its higher luminescence-signal-to-background ratio [45], which makes it suitable for luminescence signal detection in metastasis models [44,46]. The obtained mouse implantation model had multiple metastatic sites not only limited to the lungs but also present in LNs and other parts of the body. Injecting cancer cells directly into the circulation through the tail vein bypasses the process of cell invasion through the extracellular matrix and intravascular invasion. It can only study the process of tumor cell survival, retention, extravasation, micrometastatic seeding, and eventual growth into fully metastatic foci in the circulation. This cell line and the fluorescent cell line were isolated and established without monoclonal screening, but rather, a polyclonal stable transplant was applied, which preserved the heterogeneity of the tumor cells to a large extent. Experimental metastasis studies of tumor cell lines have shown that the reinjection of metastatic cell populations can lead to the enrichment of the metastatic phenotype [47,48], and advances in animal modeling, in vivo imaging, and functional genomics have accelerated the discovery of critical molecular mediators of organ-specific metastasis, which will be explored further in future.

The most common model of circulating inoculation involves injecting cancer cells into the tail vein, which is performed to assess the characteristics of metastasis after the tumor enters the vasculature. This method usually leads to pulmonary metastasis; however, it can bypass the possibility of metastasis by lymphatic spread [24]. Our established cell line, when injected into the tail vein of BALB/c-nu mice, clearly led to varying degrees of lung metastasis on day 6, and a luminescent signal at the location of popliteal LNs in the hindlimb on day 9. Our established cell line can undergo distal metastasis following tail vein injection, colonizing the LNs and lungs, and it is a better model for studying systemic therapy. Next, our lab will continue to undertake research into target and immunotherapeutic drugs at the cellular and animal levels.

## 5. Conclusions

Further research remains to be performed on the treatment and causative factors of oral melanoma in dogs. The COMM6605 melanoma cell line isolated and established in this study stably expresses specific proteins expressed in canine primary tumors and remains expressed in situ and subcutaneous implantation in mice. Xenograft mouse models established using the COMM6605 cell line are instrumental in studying mucosal melanoma progression. Additionally, two derived sublines have been successfully established to incorporate the firefly luciferase gene/ fluorescent gene, enabling the in vivo simulation of the latter steps of hematogenous spread during tumor metastasis of mucosal melanoma in mice. Given the capacities of the available technology, the whole genomes of the cell lines were not fully characterized in this study. However, this is necessary and something we have always sought to achieve. Next, we will explore the genes and transcriptome of the cell lines in more detail, as well as undertake drug trials and other research.

## Figures and Tables

**Figure 1 cells-13-00992-f001:**
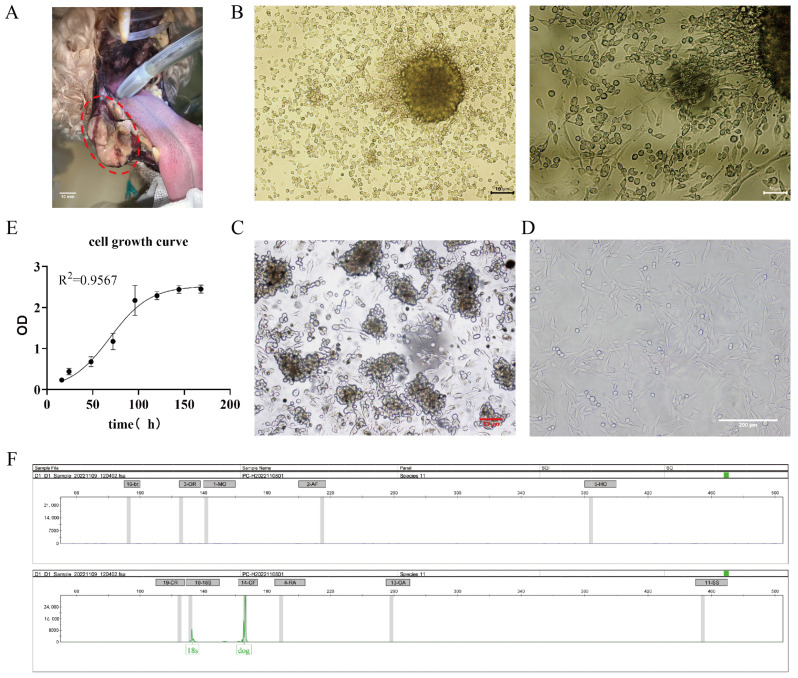
(**A**) Photograph of canine oral tumor, the area circled by the red dotted line in the figure is the oral mass; scale bar 10 mm. (**B**) Tissue block culture method day 3 microscopic observation; scale bar 100 μm and 50 μm. (**C**) Isolated cells obtained from the bacterial suspension cultured on day 2, microscopic observation; scale bar 50 μm. (**D**) Light microscopic observation of the cellular morphology (passaged to the 50th generation); scale bar 200 μm. (**E**) The growth curves of the cells. (**F**) The cellular species identification results. Capillary electrophoresis profiles of COXI gene amplification products of cell lines (endogenous reference gene: 18S; canine source-specific primer: 14-CF; the green peak is visible in the figure). The positions of the internal reference gene and amplification peaks are normal; the canine species-specific primer position appears in the form of a specific peak, and no amplification peaks can be seen in the other species-specific primer positions.

**Figure 2 cells-13-00992-f002:**
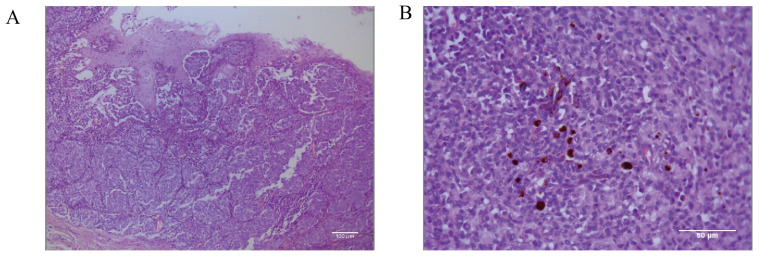
Pathological diagnostic results of canine oral tumors observed microscopically. (**A**) Histopathology of the canine oral tumor under low magnification; the junction of mucosa and submucosa contained many tumor cells, and locally, the cells were seen to have a minimal amount of melanin deposition; scale bar 100 μm. (**B**) A small number of melanin granules were clearly seen under high magnification; the tumor cells were heterogeneous, with unequal nuclei, small protruding nuclei, and mitotic divisions; scale bar 50 μm.

**Figure 3 cells-13-00992-f003:**
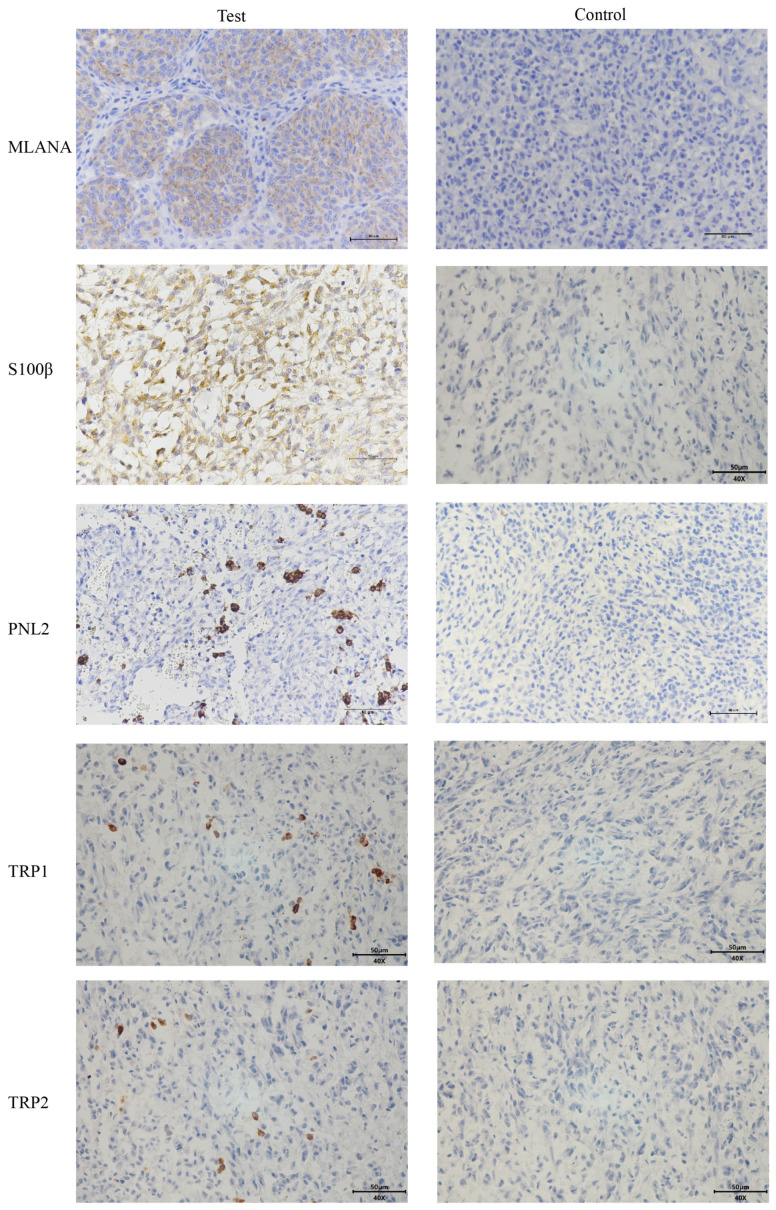
Immunohistochemical (IHC) staining results of canine oral tumors. Melan-A (MLANA), S100 beta protein (S100β), PNL2, tyrosinase-related protein 1 (TRP1), and tyrosinase-related protein 2 (TRP2) were positive; Controls were identical canine tumor tissue controls without primary antibodies; scale bar 50 μm.

**Figure 4 cells-13-00992-f004:**
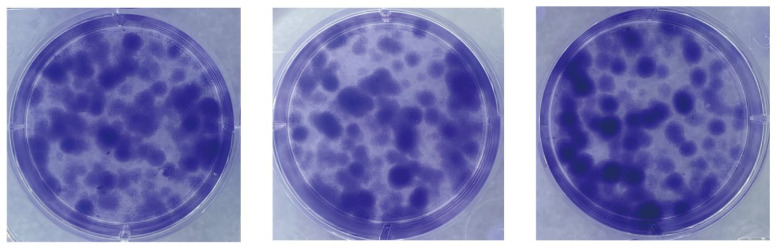
Single-cell clone formation assay. Single-cell clonal populations formed after 10 days of single-cell spread culture, as seen by staining with crystal violet. The cell community formed by each cell clone is a separate, purple-colored dot. Three replicate culture wells were made.

**Figure 5 cells-13-00992-f005:**
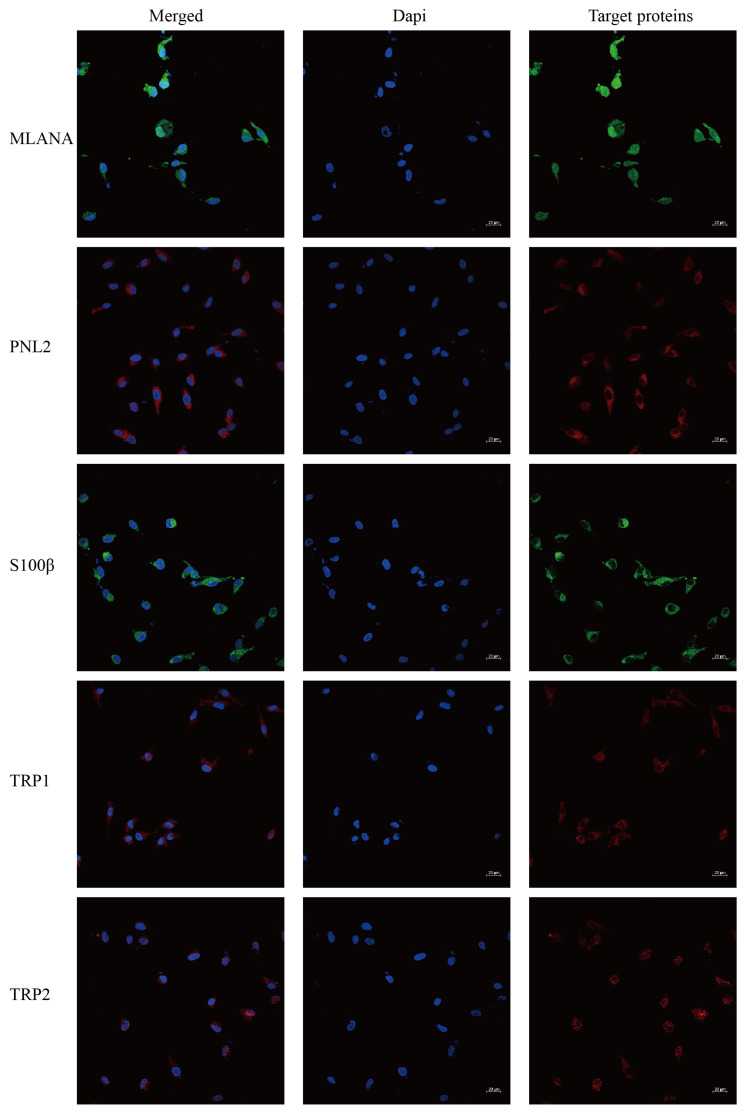
Immunofluorescence staining results of COMM6605 cells, MLANA (+), S100β (+), PNL2 (+), TRP1 (+), TRP2 (+); scale bar 20 μm.

**Figure 6 cells-13-00992-f006:**
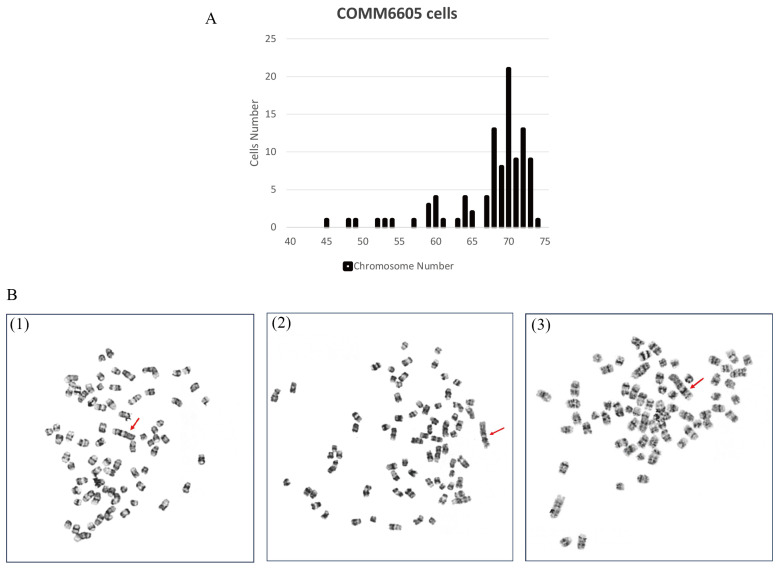
Karyotype analysis of COMM6605 tumor cells. (**A**) Abnormal chromosome number. Chromosome counting of 100 cells yielded a chromosome number range of 45~74 for COMM6605. (**B**) The metaphase of chromosomes of COMM6605 tumor cells were photographed under the microscope after using G-banding staining as shown in Figures. Figure (B1–B3) are all microscopic photographs of metaphase chromosomes in the nucleus of the cells. The red arrows point to the marker chromosome of this cell line, which are formed by the centric fusion of chromosomes 1 and 5.

**Figure 7 cells-13-00992-f007:**
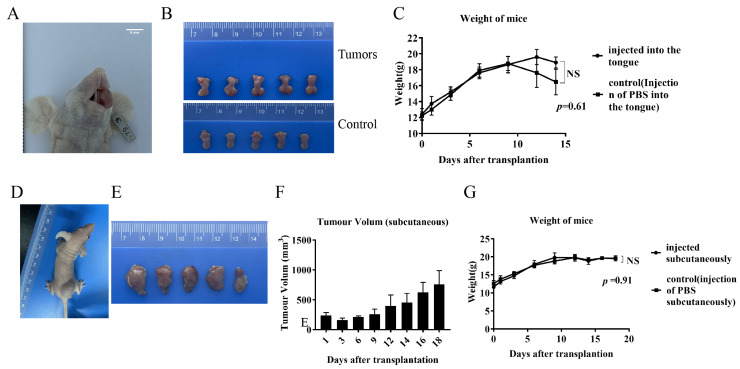
Orthotopic xenograft and subcutaneous xenograft models. (**A**) Mouse oral in situ implant model. (**B**) Tongue mass harvested from mouse oral in situ implant model and control (injected with the same volume of PBS in the tongue) normal tongue. (**C**) Body weight changes of oral-implanted tumor-bearing mice and control mice during the experimental period. (**D**) Mouse subcutaneous xenograft implant model. (**E**) Swellings harvested from subcutaneous xenograft implantation of mice. (**F**) Volume changes of subcutaneous swellings on the backs of mice during the test cycle. (**G**) Body weight changes of subcutaneously implanted mice and control mice (injected subcutaneously with the same volume of PBS) during the test period.

**Figure 8 cells-13-00992-f008:**
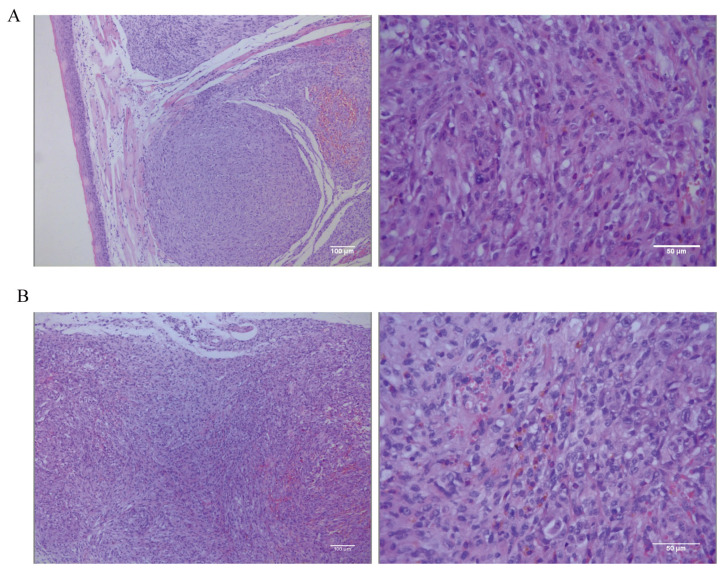
(**A**) Low-magnification (scale bar 100 μm) and high-magnification (scale bar 50 μm) hematoxylin–eosin (HE) staining results of the tongue mass of mice. (**B**) Low-magnification (scale bar 100 μm) and high-magnification (scale bar 50 μm) HE staining results in subcutaneous swelling on the backs of mice.

**Figure 9 cells-13-00992-f009:**
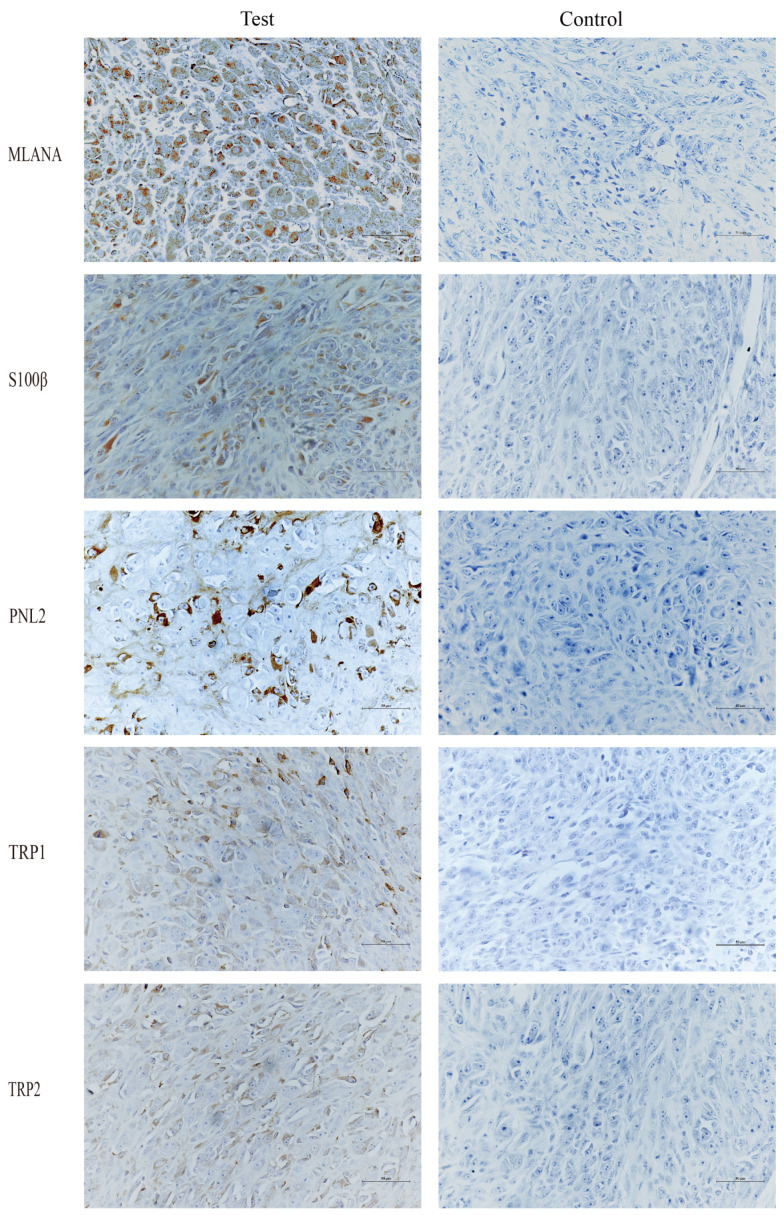
IHC results of tongue swelling in mice, MLANA (+), S100β (+), PNL2 (+), TRP1 (+), TRP2 (+); controls were identical mouse tumor tissue controls without primary antibodies. Scale bar 50 μm.

**Figure 10 cells-13-00992-f010:**
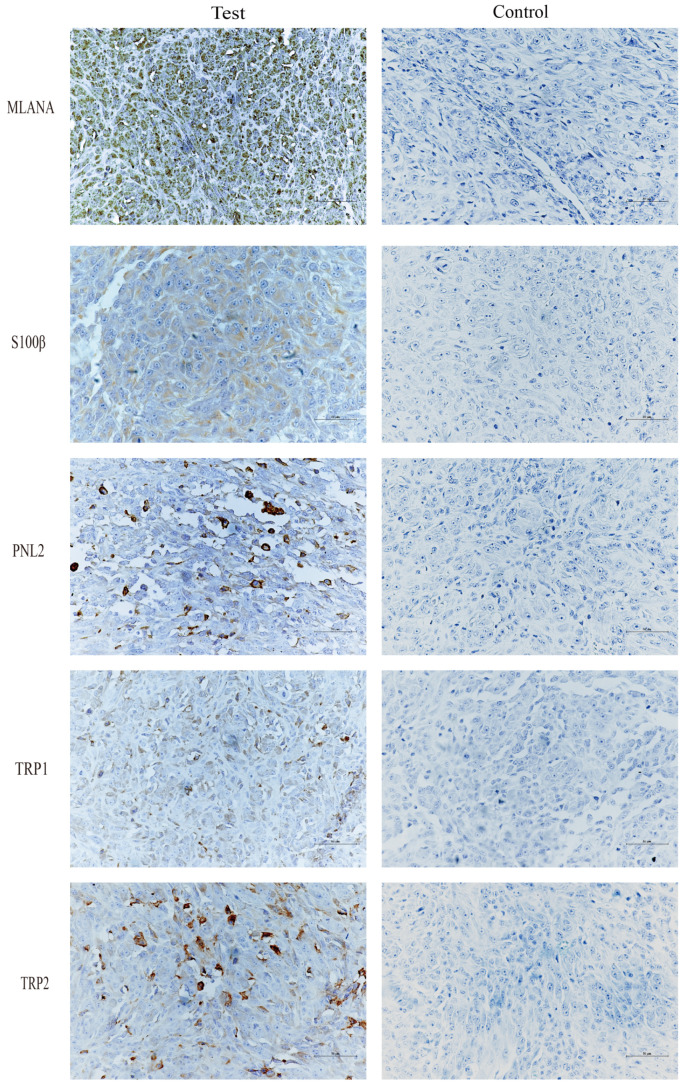
IHC results of subcutaneous swelling on the backs of mice, MLANA (+), S100β (+), PNL2 (+), TRP1 (+), TRP2 (+); controls were identical mouse tumor tissue controls without primary antibodies. Scale bar 50 μm.

**Figure 11 cells-13-00992-f011:**
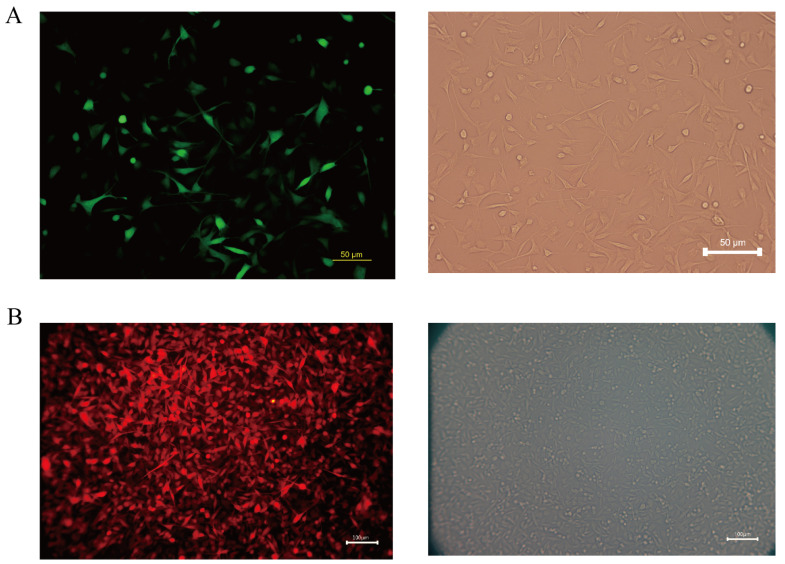
(**A**) Fluorescence (enhanced green fluorescent protein, EGFP) and white light imaging of COMM0665-Luc-EGFP; scale bar, 50 μm. (**B**) Fluorescence (Cherry fluorescent protein) and white light imaging of COMM6605-Cherry; scale bar, 100 μm.

**Figure 12 cells-13-00992-f012:**
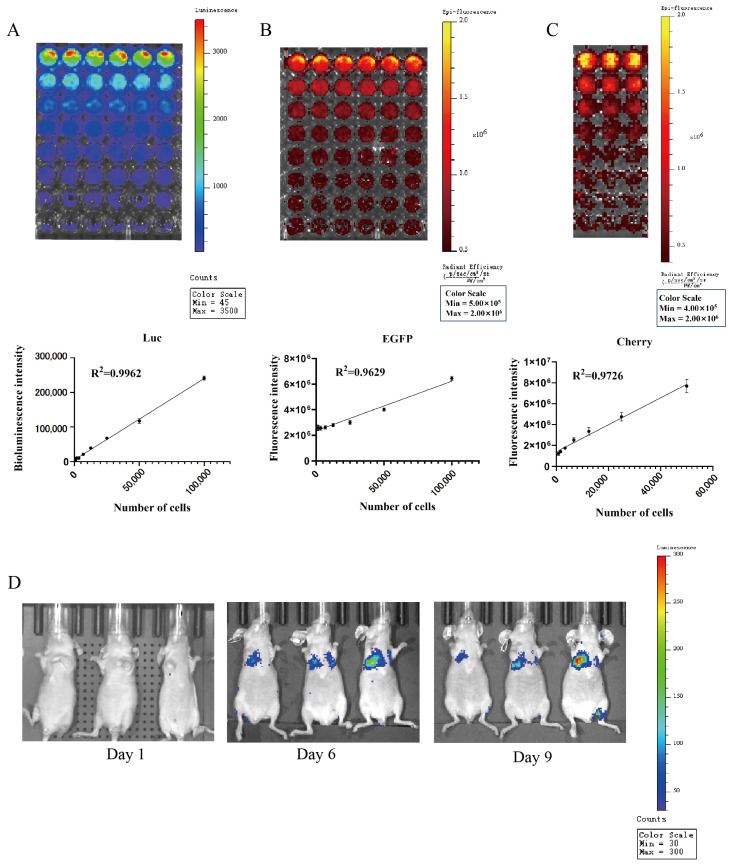
(**A**) Bioluminescence of COMM0665-Luc-EGFP cells and correlation analysis between bioluminescence intensity and cell density in the presence of firefly luciferase (Luc) after addition of D-luciferin potassium salt. Each value is the average of 6 replicate experiments, mean ± standard deviation. (**B**) Fluorescence of COMM0665-Luc-EGFP cells and correlation analysis between fluorescence intensity and cell density. Each value is the average of 6 replicate experiments, mean ± standard deviation. (**C**) Fluorescence of COMM6605-Cherry cells and correlation analysis between fluorescence intensity and cell density. Each value is the average of 3 replicate experiments, mean ± standard deviation. (**D**) Bioluminescence imaging of mice metastatic tumor models. Bioluminescence intensity of COMM0665-Luc-EGFP cells after days 1, 6, and 9 of implantation in mice were captured using the IVIS imaging system. n = 3.

## Data Availability

The original contributions presented in the study are included in the article/Appendix A; further inquiries can be directed to the corresponding author.

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
