# Peer review of "Establishment of Canine Oral Mucosal Melanoma Cell Lines and Their Xenogeneic Animal Models"

_cells, 2024, doi:10.3390/cells13110992_

Round 1

Reviewer 1 Report

Comments and Suggestions for Authors

This manuscript by Jin and colleagues described the isolation and characterization of canine oral melanoma cell lines. To show these cells are tumorigenic in vitro and in vivo, growth characteristics were examined, and xenografts were performed in immunocompromised nude mice. The cells were also tagged with lentiviral vectors with luciferase or EGFP and the small animal imaging IVIS system was used to monitor the growth and progression of the implanted tumor cells. This study is good and provided a source of canine mucosal melanoma cell lines for further investigation. There are some questions and comments shown below:

1.     Tumor tissues were removed from how many dogs? What is the success rate in establishing cell lines?  the results described here is from only one tumor? In the conclusion the authors stated there were three cell lines established? You mean one cell line and two subclones with different tags? They are not three cell lines.

2.     Please convert rpm to xg.

3.     It is unlikely dorsally implanted tumor cells will form visible tumor on day 2, the observed “tumors” are probably the inoculated cells.

4.     Tail vein injection to model metastasis is commonly used but not a real assay for metastasis, it is an assay to assess the homing of the inoculated cells to lung/liver. Also it is not realistic to have metastasis within 6 days after inoculation of cells.

5.     It is also not possible to correlate the intensities of biotin luminescence and fluorescence (Figure 8A and 8B) with the number of cells inoculated.

6.     It is surprising that the tagged tumor cells are not visible at site of injection (tail), figure 8A.

7.     It has been stated by many investigators that the use of lentiviral vector to tag cells is good with high success rate but within a few months the in vivo tagged tumor cells lose the tag thus it will be difficult to monitor the growth/progression of inoculated tumor cells.

8.     It will be very interesting to know if biochemically and/or molecularly there were any differences between orthotopic implanted and subcutaneously inoculated tumors.

9.     Pls review the English language.

Comments on the Quality of English Language

Pls check the English language, improvement is needed.

Reviewer 2 Report

Comments and Suggestions for Authors

Li et al. report development of canine mucosal melanoma cell line and its characterization in vitro and in vivo in murine model. Although this is study in relevant to the field, it has several weaknesses. The development of canine melanoma cell line isn’t that new. In fact, several canine melanoma cell lines have been reported in previous publications. In addition, manuscript is so poorly written, making it even hard to follow. The study is not fit for publication in the present format. Authors should address the following comments.

Major comments.

1.    The manuscript has been written so badly that it completely lacks any publication standard. At many places throughout the manuscript, writing is so poor that it even fails to convey a conclusion. Authors must edit manuscript for clarity and presentation.

2.    Authors analyzed several markers and used other assays for cell line authentication. Based the results of these analyses, authors concluded that their cell line is in fact a type of canine melanoma cell line. Although these results are encouraging, the data is not sufficient. Authors must complement this data with more selective assays for cell line characterization, such as karyotyping, whole exome sequencing for genetic lesions, and other similar assays.

3.    The cell line testing murine model is valuable. However, the murine model for metastatic melanoma isn’t very convincing. I do not feel tail vein injection is a right approach for examining metastasis. Authors could inoculate low number of tumor cells into the tongue, which would give more time for cells to metastasize to LN, lungs, and other sites. Alternatively, analyzing micrometastasis using IHC and other assays of tissue samples would be helpful to study their cell line metastasis.

Minor comments.

1.    Authors used the expressions such as “mice were killed”. I do not feel this is appropriate. I would suggest authors to use either “euthanized” or “sacrificed”.

2.    In results section: “Establishment of a canine oral melanoma cell line that can be stably inherited”; what do you mean my “stably inherited”.

3.    Figure 1F: Some description about this assay could be helpful to readers; such as how it differentiates species?  Whether it was used previously for similar purpose?

4.    Figure 2A, I don’t see description for images (right and left panel). What are these, low and high magnification? The scale bar in the image is not clearly visible anyway. In figure 2B, what is the “control”, not described? Is this a surrounding mouse tissue? If so, have you confirmed the cross reactivity of your antibodies to mouse and canine targets. Your controls are completely negative for most markers, that is a bit surprising.

5.    In figure 3, you have three images, what are these? Are these replicates, no description?

6.    Figure 6A, same as in figures 2A, the scale bar in the image is not clearly visible. In figure 6B, what are controls here, normal surrounding tissue of mouse. If so, have you verified if your antibodies cross react with mouse and canine targets? As in figure 2B, control tissues are completely negative for most markers, which is surprising.

7.    Authors have used commercially generated lentivirus for production of fluorescent/luc protein expressing cell lines. Authors have not provided any description on these virus particles. Are these vsv pseudotyped lentiviruses? How are these particles produced? A description on these particles helps.

8.    This discussion section is both insufficient and out of the context at some places. For example, in the first paragraph, it was not clear what authors trying to discuss here. It looks like they are describing the cell line development methodology. Discussion is not a place for such thing. Authors really need to work on the discussion section.

Comments on the Quality of English Language

See in the comment section above.
